

# Multifractality and its role in anomalous transport in the disordered XXZ spin-chain

David J. Luitz[1*], Ivan M. Khaymovich[1†] and Yevgeny Bar Lev[2∘]

**1** Max-Planck-Institut für Physik komplexer Systeme,
Nöthnitzer Str. 38, 01187 Dresden, Germany
**2** Department of Physics, Ben-Gurion University of the Negev, Beer-Sheva 84105, Israel

* dluitz@pks.mpg.de, † ivan.khaymovich@pks.mpg.de, ∘ ybarlev@bgu.ac.il

## Abstract

The disordered XXZ model is a prototype model of the many-body localization (MBL) transition. Despite numerous studies of this model, the available numerical evidence of multifractality of its eigenstates is not very conclusive due to severe finite size effects. Moreover it is not clear if similarly to the case of single-particle physics, multifractal properties of the many-body eigenstates are related to anomalous transport, which is observed in this model. In this work, using a state-of-the-art, massively parallel, numerically exact method, we study systems of up to 24 spins and show that a large fraction of the delocalized phase flows towards ergodicity in the thermodynamic limit, while a region immediately preceding the MBL transition appears to be multifractal in this limit. We discuss the implication of our finding on the mechanism of subdiffusive transport.

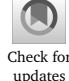

# 1   Introduction

Metal-insulator transitions are central in condensed matter physics. In most of these transitions the insulating phase is gapped and the conductivity is mediated by thermal activation across the gap. It is thus exponentially suppressed at sufficiently low temperatures strictly vanishing at absolute zero. However in the presence of strong quenched disorder, and in the absence of interactions, a different kind of metal-insulator transition is possible, which is called the Anderson localization transition [1]. Across the Anderson transition the spectrum is gapless and the transition occurs due to the change in the nature of the eigenfunctions [1]. In the metallic phase the eigenfunctions are ergodic and extended, namely the probability to find a particle in a certain position is approximately uniform in space. On the other hand, on the insulating part the eigenfunctions are localized, such that the particle is found in the vicinity of a certain point. The Anderson transition point is special, since at this point the eigenfunctions are neither ergodic and extended nor localized; they cover a sub-extensive number of sites, a situation which is called multifractality or nonergodic extended phase [2]. The spatial structure of the eigenfunctions is directly related to the transport in the system. Ergodic and extended eigenfunctions yield diffusion, while localized eigenfunctions suppress all transport all together. At the critical point, the system is known to have subdiffusive transport, with a fixed dynamical exponent [2].

Almost 15 years ago, it was shown that sufficiently weak interactions between the particles do not destroy the Anderson insulator, but induce a transition, known as the many-body localization (MBL) transition between an delocalized and localized phases [3] (see [4] for a recent review). Signatures of MBL were observed in ultracold atomic gases on optical lattices both in one-dimensional [5–7] and two-dimensional systems [8]. Similarly to the Anderson transition, the MBL transition was believed to be a *finite* temperature transition between a diffusive metal and an insulator, with the crucial difference that in the insulating phase, the conductivity of a thermodynamically large system is *strictly* zero even at finite temperature [3, 9]. A number of numerical studies demonstrated later, that for one-dimensional systems with bounded energy density, transport in the delocalized phase is *subdiffusive*, and thus conductivity in the thermodynamic limit vanishes through the entire phase diagram [10–14]. In addition to the anomalous transport, the delocalized phase shows sublinear growth of entanglement entropy [13, 15, 16], suppressed spreading of entanglement [17–19], intermediate statistics of eigenvalue spacing [20] and satisfies only a modified version of the eigenstates thermalization hypothesis (ETH) [21] (see [22] for a detailed review of the properties of the delocalized phase). A phenomenological explanation of the anomalous dynamical properties of the delocalized phase, based on rare blocking regions, was provided in Refs. [12, 23] (see also recent review [24]), however a number of predictions of this theory are not entirely consistent with numerical studies [25–27] and experiments [28] (although there is also supporting numerical evidence [29]).

Since anomalous relaxation and transport are in many cases related to multifractality of the eigenstates [30, 31], a natural question to ask is whether a similar relation exists also for the delocalized phase in systems which exhibit the MBL transition. This direction of thought is evermore suggestive, since MBL is often viewed as Anderson localization in Fock space, or more concretely on a complicated high-dimensional graph, where the nodes are Fock states, and the connectivity between them is mediated by the Hamiltonian (cf. discussion in [32]). Since the structure of this graph is rather involved it is normally approximated by either the Bethe lattice [33] or random-regular graphs (RRG, see also review by Imbrie *et al.* [34]). In addition, the disorder residing on the nodes of this graph, is highly correlated, a feature which was shown to

---

[1]The transition occurs only at three dimensions or higher for tight-binding models without spin-orbit coupling, and in any dimension for long-range random matrix models as discussed, e.g., in Refs. [2, 76, 87–90]

be important for MBL [35, 36] compared to the Anderson problem on RRG. The first proposal of an intermediate nonergodic extended phase sandwiched between the deeply ergodic and insulating (MBL) phases appeared almost 20 years ago [33]. This phase, colloquially dubbed by Altshuler a "bad metal" [37], was defined as a phase where the eigenfunctions are extended over the Hilbert space, but cover only $\mathcal{N}^\gamma$ states, where $\gamma < 1$ and $\mathcal{N}$ is the Hilbert space dimension. Whether such an intermediate phase, with multifractal eigenfunctions, exists for the Anderson localization problem on the Bethe lattice or RRGs, is still an ongoing debate. Large scale studies on random regular graphs (RRGs) suggest that this phase disappears in the thermodynamic limit [38–42], although there is also no consensus here [38–40, 43–48]. In addition, for weak disorder where all researchers agree that eigenfunctions are ergodic on RRGs, subdiffusion has been recently observed [49, 50]. Notwithstanding, while Anderson localization on graphs and MBL are related, it is not clear whether results from RRGs apply for MBL.

Multifractal properties of eigenstates of systems which exhibit MBL where examined in a number of studies [51–55]. The outcome is however rather inconclusive, mostly due to presence of severe finite size effects (mentioned as well in recent papers [56–60]). While Ref. [51] suggests that there is no intermediate multifractal phase, Refs. [52, 61] argue in favor of a stable intermediate phase. In Refs. [53, 62] multifractal properties of matrix elements of local operators are studied and found to be multifractal, though Ref. [53] argued that the intermediate phase shrinks to the MBL critical point in the thermodynamic limit, as it occurs in the standard Anderson transition. There is therefore a need for a large-scale, numerical study, which attempts to resolve these discrepancies, and shed light whether multifractality is related to the anomalous dynamical properties of the delocalized phase. Two multifractal moments of eigenstates of the disordered XXZ model were studied in Refs. [51, 54], and suggest that the extended phase is ergodic. While we see similar behavior of the relevant moments, in our work we find them insufficient to unveil possible nonergodic behavior, which becomes only apparent at higher moments. Our analysis thus allows us to locate a region in the extended phase which appears to be nonergodic within the available system sizes.[2] Our study also provides, for the first time, the presentation of the multifractal spectrum. We are able to identify a large portion of the delocalized phase, where anomalous transport was previously observed, but which is consistent with a transient multifractality. Our results support the existence of multifractality in a region which precedes the MBL transition, although we cannot say whether this region shrinks to the critical point when the system size is increased (cf. [53]).

## 2 Model

In this work we analyze the properties of the disordered XXZ chain, which is given by the Hamiltonian,

$$\hat{H} = \frac{J_{xy}}{2} \sum_{i=1}^{L-1} \left( \hat{S}_i^+ \hat{S}_{i+1}^- + \hat{S}_i^- \hat{S}_{i+1}^+ \right) + J_z \sum_{i=1}^{L-1} \hat{S}_i^z \hat{S}_{i+1}^z + \sum_{i=1}^{L} h_i \hat{S}_i^z, \tag{1}$$

where $\hat{S}_i^z$, is the $z-$projection of the spin-1/2 operator, $\hat{S}_i^\pm$ are the corresponding lowering and raising operators, $J_{xy}$ and $J_z$ are inter-spin couplings and $h_i$ are random magnetic fields taken to be uniformly distributed in the interval $h_i \in [-W, W]$. This model conserves the $z-$projection of the total spin, and serves as the prototypical model of the MBL transition, which for infinite temperature occurs for $W \sim W_c \simeq 3.7$ [51, 63, 64]. For $W \gtrsim W_c$ the system

---

[2]This cannot completely rule out strong finite-size effects mentioned in Refs. [56–60], although the presence of multifractal symmetry for the spectrum of fractal dimensions can be considered as a quite strong argument against such a scenario.

is in a MBL phase, with a completely suppressed transport of all globally conserved quantities [3], while for $W \lesssim W_c$ it exhibits an anomalous transport with a dynamical exponents which depends on the disorder strength [10–14]. We note in passing that while the value of the critical disorder $W_c$ determining the MBL transition in the XXZ Heisenberg model is under debate (cf. $W \simeq 3.7$ in Refs. [51, 63, 64] vs $W \gtrsim 4.5$ in Refs. [59, 60, 65, 66]), since one of the objectives of this work is to study the connection between anomalous transport and multifractality to avoid the controversy we limit the disorder strengths in our study to , $W \leq 3$, which according to all studies belong to the delocalized phase.

## 3  Results

Multifractal analysis requires the calculation of the eigenstates of (1) in a certain energy density window and for a large number of disorder realizations. Since full diagonalization becomes overwhelmingly expensive for system sizes $L \gtrsim 18$, and access to large system sizes is essential, we utilize the shift-invert technique [67], which transforms the spectrum of the Hamiltonian such that the states of interest are moved to the lowest (highest) energies in the transformed spectrum and become tractable by Krylov space methods. The most commonly used spectral transformation for this purpose is $(H - \sigma I)^{-1}$, where the explicit inversion of the shifted Hamiltonian can be avoided and replaced by the solution of a set of linear equations using the Gauss algorithm. We use the massively parallel `strumpack` library [68, 69] to extract about 50 eigenstates in the middle of the many-body spectrum, where the density of states is at its maximum. The largest system size we consider is $L = 24$, which corresponds to a Hilbert space dimension of $\mathcal{N} = 2\,704\,156$. We repeat this procedure for $100 - 15\,000$ realizations of the disordered magnetic field $h_i$ in (1). Overall, the total number of calculated eigenstate coefficients in the computational basis for each disorder strength and system size is $10^8 - 10^{10}$, which in most cases allows us to reach statistical errors smaller than the symbol size.

### 3.1  Distributions of eigenstate coefficients

The high energy states of *ergodic* systems are well approximated by eigenstates of random-matrices drawn from a Wigner-Dyson ensemble of matrices [70] which shares the same temporal symmetry as the Hamiltonian. Specifically, eigenstates of real ergodic Hamiltonians, which are time-inversion invariant, are well described by eigenstates of matrices drawn from the Gaussian Orthogonal Ensenble (GOE), suggesting that the elements of eigenstates, $|\beta\rangle$, written in a certain basis $|n\rangle$, are almost independent random variables, normally distributed according to,

$$P_{\text{GOE}}(x \equiv |\langle n|\beta\rangle|) = \sqrt{\frac{\mathcal{N}}{2\pi}} e^{-\mathcal{N}x^2/2}, \tag{2}$$

where we defined the random variable, $x \equiv |\langle n|\beta\rangle|$ and $\mathcal{N}$ is the Hilbert space dimension. This assertion, known as Berry's conjecture [71], was verified numerically in several single- and many-body ergodic systems (see Ref. [72] for a review). Multifractal eigenstates, on the contrary, do *not* satisfy Berry's conjecture, but are distributed according to,

$$P_{\mathcal{N}}(x) \propto \frac{1}{|x|} \mathcal{N}^{f\left(-\ln x^2/\ln \mathcal{N}\right)-1}, \tag{3}$$

where $f(\alpha)$ is a function called the spectrum of fractal dimensions [2], depending on the only variable $\alpha$ taken to be $\alpha \equiv -\ln x^2/\ln \mathcal{N}$ (see Section 3.3, and Eq. (14) for the form of $f(\alpha)$ for GOE eigenstates).

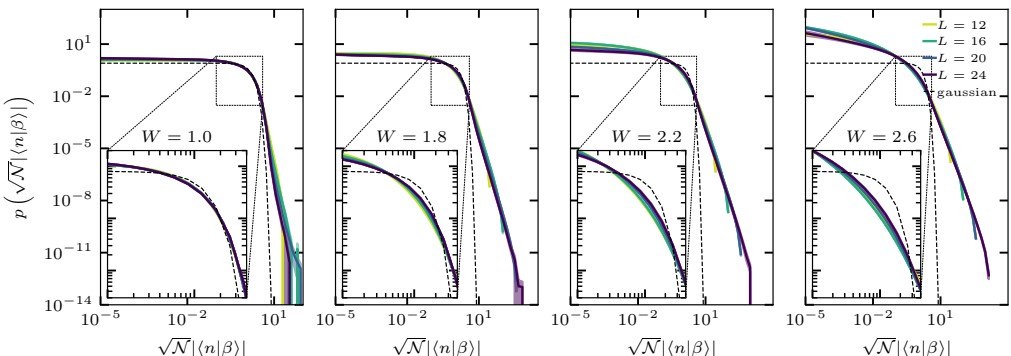

Figure 1: Normalized probability density of scaled eigenstate coefficients in the computational basis, $P\left(\sqrt{\mathcal{N}}\,|\langle n|\beta\rangle|\right)$ for disorder strengths $W = 1.0, 1.8, 2.2$ and $2.6$ and system sizes $L = 12, 16, 20$ and $24$ (larger systems correspond to darker colors). The dashed black line represents the normal distribution and errorbars are represented by shaded areas of the order of the line width.

The distribution of eigenstate coefficients for our model (1) has been studied by two of us in Ref. [21], and was found to exhibit significant deviations from Berry's conjecture for $0.4 \le W \le 1.8$, hinting that the underlying eigenstates are multifractal. In this work we study these distributions in detail, focusing on their flow towards the thermodynamic limit. In both above cases we focus on eigenstate coefficients in the computational basis, where the basis states $|n\rangle$ are labeled by the eigenvalues of the local $\hat{S}_i^z$ operators.

To give equal weight to small and large values of the eigenstate coefficients, the bins of the histogram are equally spaced on a *logarithmic* scale, which we achieve by calculating the histogram of $\alpha \equiv -\ln|\langle n|\beta\rangle|^2/\ln\mathcal{N}$ using bins of equal size. The histogram of the wavefunction coefficients $|\langle n|\beta\rangle|$ and the corresponding probability density can then be straightforwardly inferred. In Fig. 1 we show the result of this calculation for a number of disorder strengths in the extended phase, and a range of system sizes. We compare these distributions with the normal distribution of GOE, (2), and see a visible departure for all disorder strengths, similarly to Fig. 2 in Ref. [21]. The departure is especially apparent in the head of the distribution, indicating an excess in small values of the eigenstate elements compared to GOE, and the tails of the distribution, indicating an excess in large values of the eigenstate elements. The departure becomes more prominent with the strength of the disorder.

While at first glance the rescaled distributions look collapsed, a more detailed examination by zooming into various parts of the distribution, shows a noticeable, yet slow, flow towards the (Gaussian) GOE distribution in most parts of the distribution. In what follows we examine this flow in detail, by considering the moments of the distribution and its multifractal spectrum (3).

## 3.2 Moments of the distributions: inverse participation ratios

In the multifractal analysis one defines the standard inverse participation ratio (IPR) $I_2^\beta$ and its generalizations,

$$I_q^\beta = \sum_n |\langle\beta|n\rangle|^{2q} \sim \mathcal{N}^{-\tau_q}, \tag{4}$$

which measure how many "sites" in the Hilbert space (a site here is a certain basis state $|n\rangle$) the wavefunction occupies [2], the generalized IPR is directly related to the corresponding $q$ Rényi entropies $S_q = \ln I_q^\beta/(1-q)$ [73]. For eigenstates extended over the entire basis, such as for eigenstates extracted from GOE, $x \equiv |\langle\beta|n\rangle| \sim \mathcal{N}^{-1/2}$ giving, $I_q^\beta \sim \sum_n \mathcal{N}^{-q} \sim \mathcal{N}^{-(q-1)}$

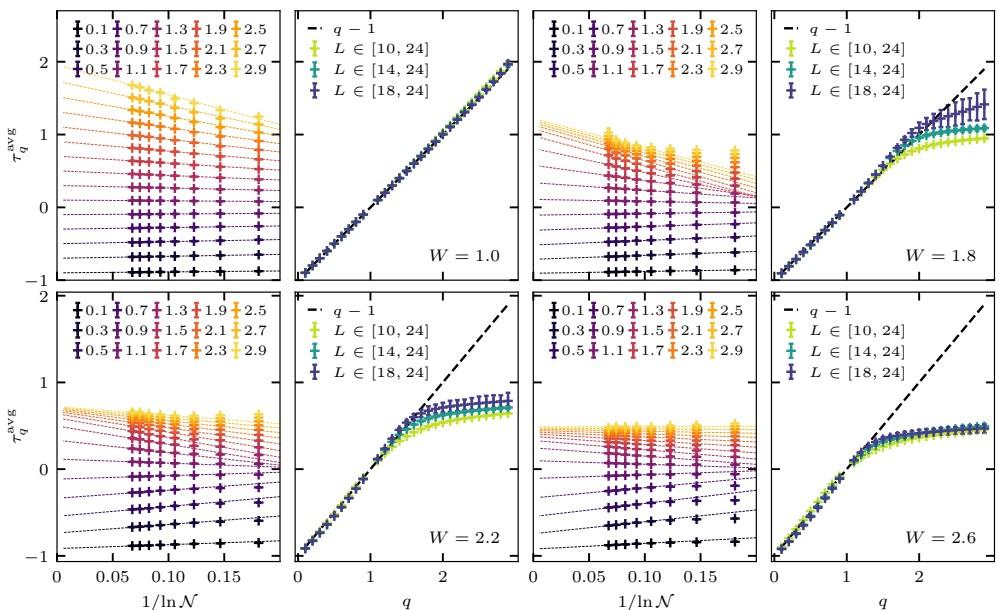

Figure 2: *Odd columns.* Finite size $\tau_q^{\mathcal{N},\mathrm{avg}}$ data (6) as a function of $1/\ln\mathcal{N}$ for various values of $q$ ($0 < q < 3$, darker shade of green indicates smaller $q$) and disorder strengths $W = 1.0, 1.8, 2.2$ and $2.6$. Statistical errors are smaller than symbol size in all plots. Dashed lines are extrapolation of the data to $\mathcal{N} \to \infty$ using a linear function in $1/\ln\mathcal{N}$. *Even columns.* Extrapolated $\tau_q^{\mathrm{avg}}$ as a function of $q$ and various extrapolation ranges indicated in the legend. Darker colors indicate more weight to larger system sizes. The dashed black line indicates $q-1$, which corresponds $\tau_q^{\mathrm{avg}}$ of the GOE ensemble.

and $\tau_q = q-1$ for $q > -0.5$ (the average of $I_q^\beta$ diverges otherwise). Eigenstates which occupy a finite number of configurations $|n\rangle$ which doesn't scale with the Hilbert space dimension $\mathcal{N}$ will have $\tau_q = 0$ for $q > 0$ (and $-\infty$ otherwise). The parameter $q$ is used to tune the weight in the average from large to small values. Under the assumption that the eigenstate coefficients are statistically independent, the IPRs are related to the moments of $P(x)$, since one can write,

$$I_q^\beta = \sum_n |\langle\beta|n\rangle|^{2q} = \mathcal{N}\frac{1}{\mathcal{N}}\sum_n |\langle\beta|n\rangle|^{2q} \approx \mathcal{N}\left\langle|x|^{2q}\right\rangle = \mathcal{N}\int |x|^{2q} P(x)\,\mathrm{d}x. \tag{5}$$

Using the definition of $\tau_q$ in (4), the normalization of the wavefunction, which gives, $I_{q=1}^\beta = 1$, and the fact that $\sum_n 1 = \mathcal{N}$, which gives $I_{q=0}^\beta = \mathcal{N}$ one can show that in the limit of $\mathcal{N} \to \infty$, $\tau_q$ is a monotonically increasing and concave function of $q$, namely $\tau_q' > 0$ and $\tau_q'' < 0$ [2].

To evaluate the $\tau_q$ we calculate the IPRs for each eigenfunction and a range $0 \le q \le 4$. We then average $I_q^\beta$ over the nearby in energy eigenstates, as also different disorder realizations, and obtain $\left\langle I_q^{\mathcal{N}}\right\rangle$. The finite-size *average* $\tau_q^{\mathrm{avg}}$ is the given by,

$$\tau_q^{\mathrm{avg}}(\mathcal{N}) \equiv -\frac{\ln\left\langle I_q^{\mathcal{N}}\right\rangle}{\ln\mathcal{N}}. \tag{6}$$

Since the relation (4) is only expected to hold asymptotically, in Fig. 2 we plot $\tau_q^{\mathrm{avg}}(\mathcal{N})$ vs $1/\ln\mathcal{N}$ and extrapolate to $\mathcal{N} \to \infty$ using a linear function in $1/\ln\mathcal{N}$ [3]. The extrapolated

---

[3]The motivation of the extrapolation versus $1/\ln\mathcal{N}$ originates from the main subleading contributions to $\tau_q(\mathcal{N})$

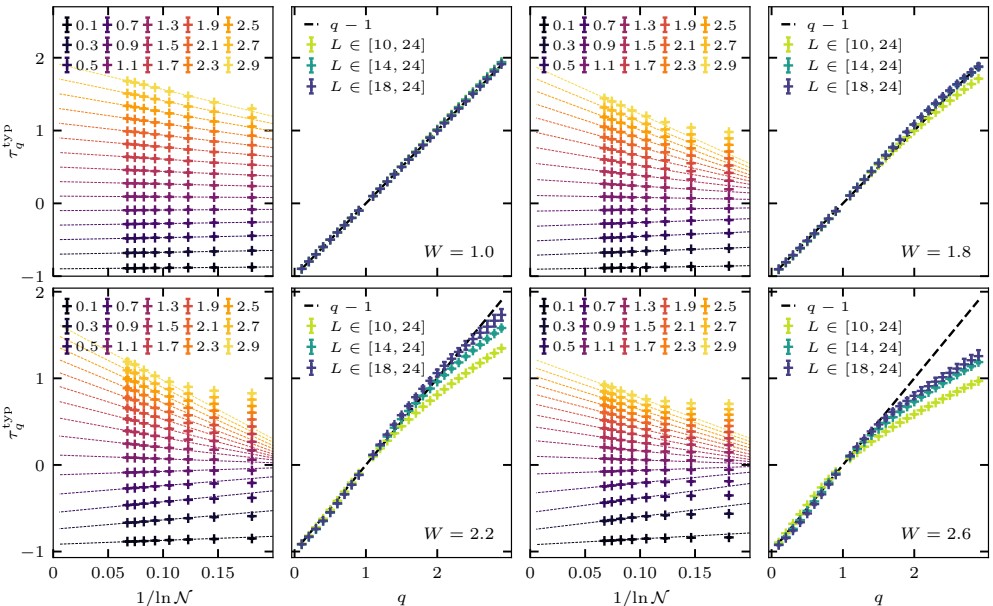

Figure 3: Similar to Fig. 2, but for $\tau_q^{\text{typ}}$, computed from Eq. (7).

values are then plotted as a function of $q$ and compared to the prediction of GOE, $\tau_q = q - 1$ (dashed black line). While the scaling of $\tau_q^{\text{avg}}(\mathcal{N})$ with respect to $1/\ln \mathcal{N}$ is mostly linear indicating a high quality of the extrapolation, for larger values of $q$ a departure from the linear dependence is apparent, suggesting that the data is still far from being asymptotic. The slight non-concavity of the extrapolated $\tau_q^{\text{avg}}$, is a finite size effect and is well within the error bars of the extrapolation. To quantify the curving of the data, we extrapolate to $\mathcal{N} \to \infty$ using a sliding fit window of system sizes, which are shown in the legend. The error bars in the extrapolated data are estimated using a bootstrap fitting procedure, quantifying the statistical errors in $\tau_q^{\text{avg}}(\mathcal{N})$ (which are in all cases smaller than the symbol size). From the extrapolated data we see that the average, $\tau_q^{\text{avg}} \sim q - 1$ for all values of $q$ up to some $q_*(W, L)$, which depends on the strength of the disorder. While for weak disorder $q_*$ spans the entire range of $q$ considered here, for $W \to W_c$ we see that $q_* \to 1$. On the other hand we also see that, $q_*(W, L)$ increases when higher weight in the extrapolation is given to the larger system sizes, suggesting that the departure from the $q - 1$ could be a finite size effect, though we cannot rule out a saturation of the form $\lim_{L \to \infty} q_*(W, L) = q_*(W)$, which will indicate residual multifractality at large moments $q > q_*(W)$.

We also study the typical $\tau_q^{\text{typ}}(\mathcal{N})$, which is defined as

$$\tau_q^{\text{typ}}(\mathcal{N}) \equiv -\frac{\ln \left\langle I_q^{\mathcal{N}} \right\rangle_{\text{typ}}}{\ln \mathcal{N}}, \qquad \text{where} \quad \left\langle I_q^{\mathcal{N}} \right\rangle_{\text{typ}} \equiv \exp\left[\left\langle \ln I_q \right\rangle\right]. \tag{7}$$

The advantage of this measure is that it suppresses the weight of the outliers. The results of the same analysis for $\tau_q^{\text{typ}}$ as described above for $\tau_q^{\text{avg}}$ are presented in Fig. 3, and are qualitatively identical to the analysis of $\tau_q^{\text{avg}}$. Quantitatively $q_*$ is pushed to much larger values [4], almost entirely eliminating the departure from the $q - 1$ line for all $W < 2.6$. This can be viewed as another indication that $q_*$ is dominated by outliers and is likely to flow to infinity for larger system sizes.

from the prefactor in (4).

[4] as in the GOE case at finite size, where $q^* \sim \ln \mathcal{N}$, see [75] for details.

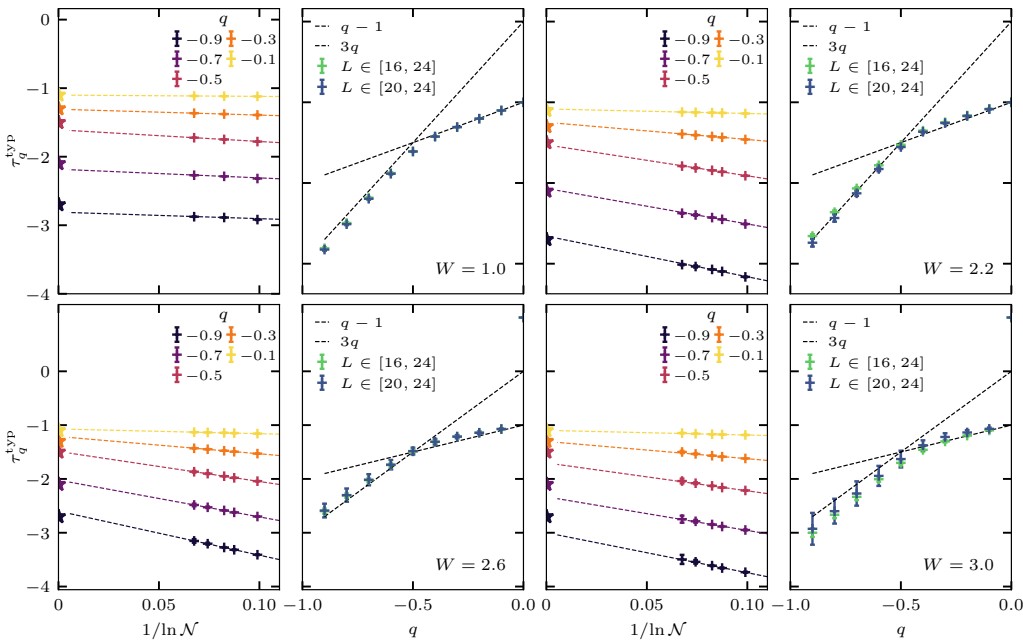

Figure 4: Similar to Fig. 3 but for $W = 1.0, 2.2, 2.6$ and $3.0$ for $q < 0$. The dashed black lines correspond to the expected behavior of $\tau_q^{\text{typ}}$ for the GOE case, $3q$ for $q < -0.5$ and $(q-1)$ otherwise. See the Eq. (9) for details.

Another advantage of $\tau_q^{\text{typ}}$ is that unlike $\tau_q^{\text{avg}}$ it doesn't diverge for $q < -0.5$, and thus allows to study the behavior of the small values of the eigenstate coefficients. We recall that these values are of particular interest given their abundance compared to the Gaussian distribution (see Fig. 1) . In Fig. 4 we repeat the analysis done in Fig. 3 for $q < 0$ (for technical reasons we use a different set of data here, which includes less samples).

We estimate the value of $\tau_q^{\text{typ}}$ for GOE eigenstates based on the behavior of $f_{\text{avg}}^{\text{GOE}}(\alpha)$ for $\alpha > 1$, which can be calculated analytically based on (2) and (3),

$$f_{\text{av}}^{\text{GOE}}(\alpha) = \begin{cases} -\infty & \alpha < 1 \\ (3-\alpha)/2 & \alpha > 1 \end{cases}. \tag{8}$$

Since the typical $f_{\text{typ}}(\alpha)$ is determined by histogram counts growing with the system size [2], it coincides with $f_{\text{avg}}(\alpha)$ for $f_{\text{avg}}(\alpha) \geq 0$ and tends to $-\infty$ (zero counts) otherwise. Thus, we can evaluate $\tau_q^{\text{typ}}$ from $f_{\text{av}}^{\text{GOE}}(\alpha)$ using a truncated Legendre transform [2],

$$\tau_q^{\text{typ}} = \sup_{\substack{\alpha \\ f_{\text{av}}^{\text{GOE}}(\alpha)>0}} \left[ q\alpha - f_{\text{av}}^{\text{GOE}}(\alpha) \right] = \begin{cases} q-1 & q > -0.5 \\ 3q & q \leq -0.5 \end{cases}, \tag{9}$$

which is designated in Fig. 4 by the dashed black lines. We note that similarly to $q > 0$, $\tau_q^{\text{typ}}$ for $q < 0$ appears to flow to the predictions of GOE.

This conclusion is in apparent contradiction to the behavior of the distributions in Fig. 1, where an excess of zeros of the eigenstates compared to GOE prediction is clearly visible for $W \sim 1$, and does not appear to vanish in the $\mathcal{N} \to \infty$. The discrepancy must follow from the prefactor in the definition of $\tau_q^{\text{typ}}$, $\left\langle I_q^{\mathcal{N}} \right\rangle_{\text{typ}} = A(\mathcal{N})\mathcal{N}^{-\tau_q^{\text{typ}}}$, which includes a slowly varying prefactor (the variation is at most of the order of $\ln \mathcal{N}$). To test this assertion numerically, we compute the ratio, $\left\langle I_q^{\mathcal{N}} \right\rangle_{\text{typ}} / \left\langle I_q^{\mathcal{N},\text{GOE}} \right\rangle_{\text{typ}}$ for a number of $q$-s and disorder strengths. Since to

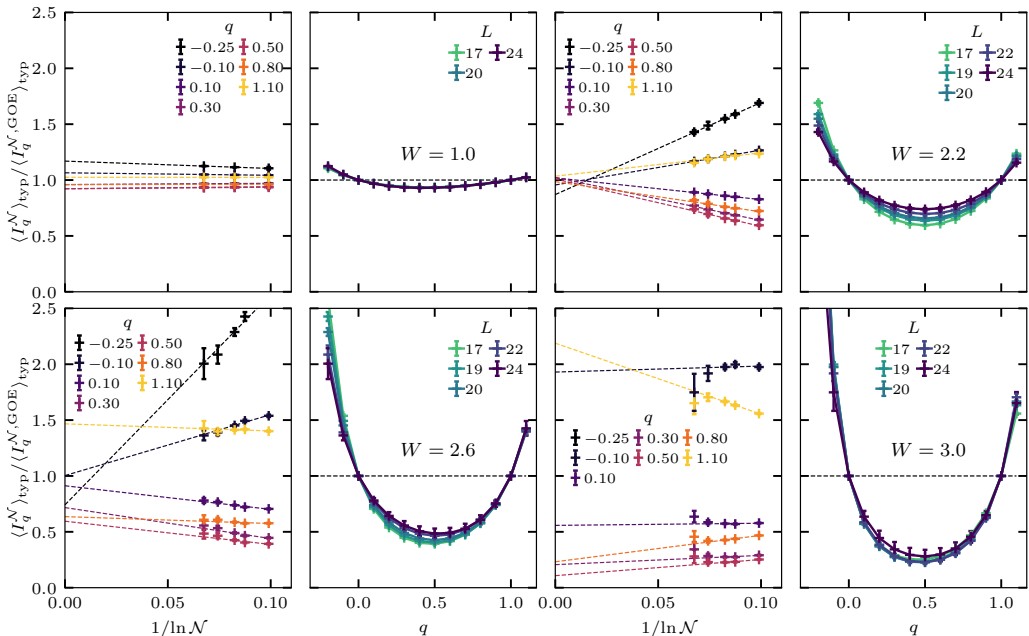

Figure 5: *Odd columns.* Finite size data of the ratio, $\left\langle I_q^{\mathcal{N}} \right\rangle_{\text{typ}} / \left\langle I_q^{\mathcal{N},\text{GOE}} \right\rangle_{\text{typ}}$ vs $1/\ln \mathcal{N}$ for a number of $q$-s and disorder strength $W$ (see legend). Dashed lines are extrapolation to $\mathcal{N} \to \infty$. *Even columns.* The same ratio $\left\langle I_q^{\mathcal{N}} \right\rangle_{\text{typ}} / \left\langle I_q^{\mathcal{N},\text{GOE}} \right\rangle_{\text{typ}}$, but as a function of $q$. Dashed black line indicates GOE limit.

the best of our knowledge there are no analytical relations for $\left\langle I_q^{\mathcal{N},\text{GOE}} \right\rangle_{\text{typ}}$ [5] we compute it numerically by drawing 100 random eigenstates from a Gaussian probability distribution in Eq. (2), while fixing the norm of the eigenstate. This procedure is very efficient and allows us to study the same Hilbert space dimensions as we do for the XXZ model, at a negligible computational cost. The results of the evaluation of $\left\langle I_q^{\mathcal{N}} \right\rangle_{\text{typ}} / \left\langle I_q^{\mathcal{N},\text{GOE}} \right\rangle_{\text{typ}}$ can be seen in Fig. 5. For $W < 1$ we indeed see that for the system sizes we have the ratio flows away from 1. We strongly suspect that this apparent divergence from GOE, is a finite size effect, which has to do with the proximity of an integrable point (for $W = 0$ the XXZ model is integrable). We leave the examination of this effect to future studies. For $W > 1$ we see an apparent flow towards 1, with clearest evidence for $W = 2.2$. For $W = 2.6$ and $3.0$, the finite size behavior is non-monotonic (highlighting the importance of the use of large system sizes), and appears to flow towards 1, though for these disorder strengths it is less apparent.

To summarize this section, we have seen that while a naïve examination of the distributions of the eigenstates elements in Fig. 1, shows apparent convergence to a non-Gaussian, and thus multifractal distribution, a more detailed analysis looking on the moments of the distribution, shows a slow but clear flow towards the predictions of GOE, in $\tau_q^{\text{typ}}, \tau_q^{\text{avg}}$ and even directly in the finite size generalized IPRs compared to thier GOE values $\left\langle I_q^{\mathcal{N}} \right\rangle_{\text{typ}} / \left\langle I_q^{\mathcal{N},\text{GOE}} \right\rangle_{\text{typ}}$. In the next section we will complement this analysis, by examining an additional multifractal measure — the multifractal spectrum.

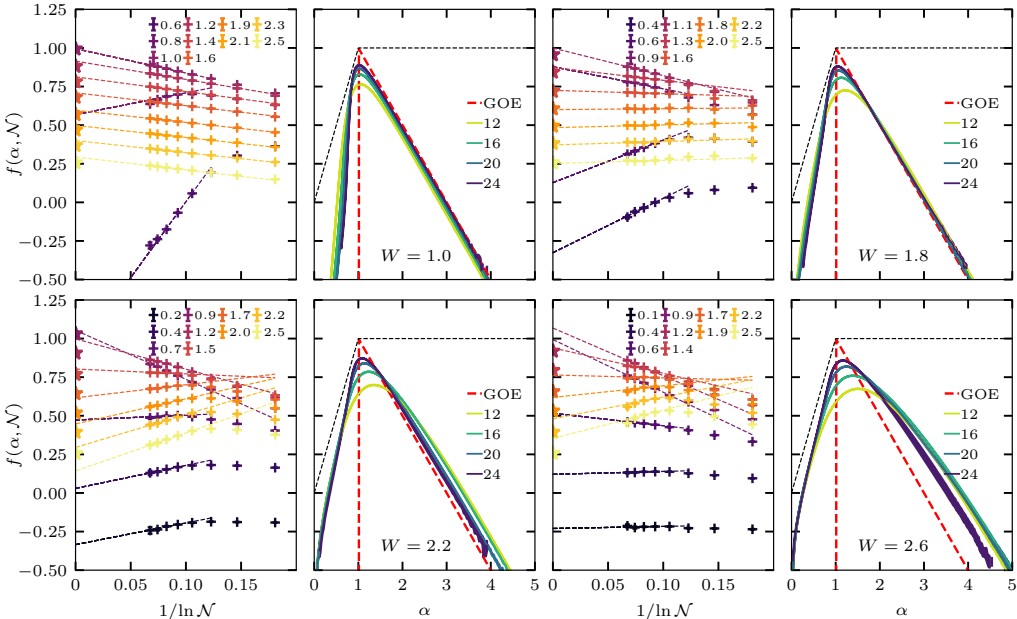

Figure 6: *Odd columns*. $f(\alpha,\mathcal{N})$ vs $1/\ln\mathcal{N}$ for different $\alpha$ values (see legend). Dashed colored lines correspond to an extrapolation to $\mathcal{N}\to\infty$, and the stars on the y-axis correspond to the infinite size GOE prediction. *Even columns*. The finite size multifractal spectrum, $f(\alpha,\mathcal{N})$ calculated using Eq. (15) for $L = 12, 16, 20, 24$ (darker colors, correspond to larger system sizes) and disorder strengths $W = 1.0, 1.8, 2.2, 2.6$, the width of the lines correspond to statistical errors. Red dashed line corresponds to the infinite size GOE prediction, $f_{\mathrm{GOE}}(\alpha)$ according to (14), and black dashed lines show the upper bounds on $f(\alpha)$ according to (13). The error bars are represented by filled areas of the order of the line width.

## 3.3 Multifractal spectrum

In this section we analyze the multifractal spectrum, $f(\alpha)$, of the eigenstates of (1), which appeared in (3), but we repeat it here for convenience,

$$P_{\mathcal{N}}(x) \propto \frac{1}{|x|}\mathcal{N}^{f\left(-\ln x^2/\ln\mathcal{N}\right)-1}, \tag{10}$$

with $\alpha \equiv -\ln x^2/\ln\mathcal{N}$ and $x \equiv |\langle n|\beta\rangle|$, where $|n\rangle$ is a basis state, and $|\beta\rangle$ are eigenstates of (1) [2]. The multifractal spectrum, $f(\alpha)$ is the fractal dimension of the set $x \in \left[\mathcal{N}^{-\alpha/2}, \mathcal{N}^{-(\alpha+d\alpha)/2}\right]$, namely the probability for $x$ to be in this interval is given by,

$$p(\alpha) \sim (\ln\mathcal{N})\mathcal{N}^{-(1-f(\alpha))}. \tag{11}$$

Using (10) and the relation (5) to $I_q$ one can see that,

$$\tau_q = \inf_{\alpha}[q\alpha - f(\alpha)], \tag{12}$$

namely in the limit $\mathcal{N}\to\infty$, $\tau_q$ and $f(\alpha)$ are related via a Legendre transform [2]. Here inf is an infimum. We note however that while $\tau_q$ is a concave function the definition (10) above allows for $f(\alpha)$ to be *non-concave* a feature which we will utilize in our analysis below.

---

[5]Unlike for the average $\left\langle I_q^{\mathcal{N},\mathrm{GOE}}\right\rangle$, see, e.g., [91].

Similarly to the restrictions on $\tau_q$, described above Eq. (5), from normalization of the probability distribution, $\int P_{\mathcal{N}}(x)\,dx = 1$ and the wavefunction $\int x^2 P_{\mathcal{N}}(x)\,dx = \mathcal{N}^{-1}$ in the limit $\mathcal{N} \to \infty$ one can derive, that

$$f(\alpha) \le \min(1, \alpha). \tag{13}$$

For the Gaussian distribution of GOE eigenstates (2), using the definition (8), one can obtain the finite size correction,

$$f_{\text{GOE}}(\alpha, \mathcal{N}) = 1 + \frac{\ln(P(\alpha)/(A\ln\mathcal{N}))}{\ln\mathcal{N}} = 1 + \frac{1-\alpha}{2} - \frac{\mathcal{N}^{1-\alpha}}{2\ln\mathcal{N}} - \frac{\ln A}{\ln\mathcal{N}}, \tag{14}$$

which in the limit $\mathcal{N} \to \infty$ gives the already mentioned result (8). Here $A$ is a normalization constant being a slow (at most logarithmic) function of $\mathcal{N}$. Note that this multifractal spectrum differs from $f_{\text{GOE}}(1) = 1$, $f_{\text{GOE}}(\alpha \ne 1) \to -\infty$ in Ref. [2] because the latter is written for wavefunction envelopes, while the raw numerical eigenstates contain de Broglie-like oscillations corresponding to the increased statistics of zeros (large values of $\alpha > 1$). While there are methods to remove these superfluous zeros (see, e.g., [45, 46]), since the statistics of the zeros does not affect $q > 0$ moments of the eigenfunctions we don't consider such methods in this work.

To obtain the finite size multifractal spectrum we numerically compute a histogram of $\alpha$ with $0 \le \alpha \le 4$ (we used 50, 100, and 200 bins, and verified that our results don't change with respect to the bin number, not shown), then using (11) yields,

$$f(\alpha, \mathcal{N}) = 1 + \frac{\ln(p(\alpha)/\ln\mathcal{N})}{\ln\mathcal{N}}, \tag{15}$$

which is presented in the even columns of Fig. 6, for various disorder strengths $1 \le W \le 2.6$. Here we assumed $A$ to be a constant (not a logarithmic function of $\mathcal{N}$) like in the GOE case as we focus on the small disorder amplitudes. In the odd columns of Fig. 6 we extrapolate the data to $\mathcal{N} \to \infty$, with the same procedure used in several random matrix models (see, e.g., [45, 46]). For sufficiently strong disorder or $\alpha$ sufficiently far from 1, our finite size data shows a nonlinear behavior in $1/\ln\mathcal{N}$ (like in GOE case (8)) indicating the importance of using large system sizes in the the determination of the asymptotic behavior. Even with the state-of-the-art system sizes we use here, the extrapolation procedure is not justified due to nonlinearity of the data for some values of $\alpha$. Nevertheless, for $W < 2$, the extrapolation works fairly well, and similarly to the moments analysis in the previous section, supports a flow towards the predictions of GOE. The extrapolation is not entirely satisfactory for $W = 2.2$ and 2.6, thus we cannot rule out multifractal behavior in this case.

We also note that in the calculation of the histograms of $\alpha$, the sampling for our system sizes starts yielding zero counts (for all used bin sizes) for $\alpha \gtrsim 2.5$ and the majority of eigenvectors. In this interesting regime (corresponding to the excess of zeros in the wavefunction histogram, Fig. 1), the (low probability) contribution to the histogram seems to stem from the distribution over disorder realizations, rather than from representative eigenstates. This leads to large fluctuations, also visible in the errorbars (shaded area).

The finite size behavior of the multifractal spectrum, $f(\alpha, \mathcal{N})$, is also useful to understand the deviation from the $(q-1)$ GOE line for both $\tau_q^{\text{avg}}$ and $\tau_q^{\text{typ}}$, which occurs for some value $q_*(W, L)$ (see Figs. 2 and 3). By looking on the direction of the flow of $f(\alpha, \mathcal{N})$ with the system size (see Fig. 6), for all $W \le 2.2$ and $\alpha$ close to the maximum of $f(\alpha, \mathcal{N})$, which corresponds to $q < q^*$, the $f(\alpha, \mathcal{N})$ *increases* with $\mathcal{N}$. For small $\alpha$ on the other hand, which corresponds to $q > q^*$, the spectrum $f(\alpha, \mathcal{N})$ *decreases* with $\mathcal{N}$. Moreover the crossing points of $f(\alpha, \mathcal{N})$ at two adjacent $\mathcal{N}$ values flow towards $\alpha = 1$ with increasing $\mathcal{N}$. For $W = 2.6$ the situation is drastically different, since there is no downward finite-size flow at small $\alpha$, but instead $f(\alpha, \mathcal{N})$ appears to saturate. This is also visible in the extrapolation curves for small

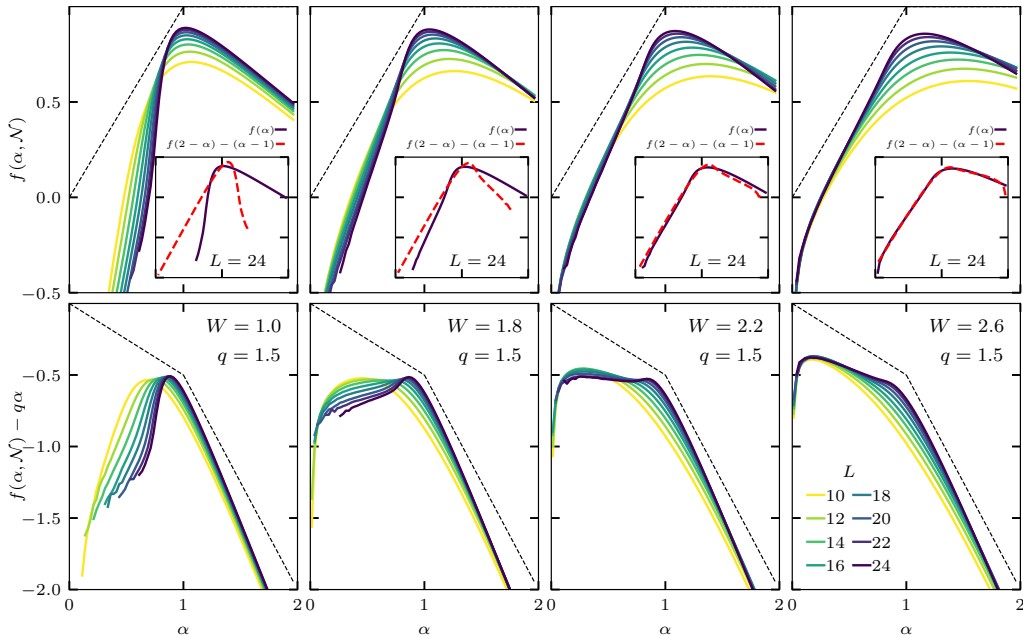

Figure 7: *Upper row*. The finite size multifractal spectrum $f(\alpha, \mathcal{N})$ for small values of $\alpha$ and $L = 12, 16, 20, 24$ (darker colors, correspond to larger system sizes). Dashed black lines indicate the upper bounds on $f(\alpha)$ according to (13). The insets show $f(\alpha, \mathcal{N})$ (solid black lines) for the maximal available size ($L = 24$) together with its multifractally symmetric counterpart according to Eq. (16) (dashed red lines). *Lower row*. Same as the upper row, but for the tilted multifractal spectrum, $f(\alpha, \mathcal{N}) - q\alpha$, and $q = 1.5$.

$\alpha = 0.1$ in Fig. 6. To emphasize this point in Fig. 7 we plot $f(\alpha) - q\alpha$, the supremum of which corresponds to $-\tau_q$ (see (9) for example). For $W = 2.6$ the left *local* maximum at $\alpha \simeq 0.1$ does not appear to flow with system size, while the *right* local maximum at $\alpha \simeq \alpha_0$ drifts upward. Nevertheless this upward flow is bounded from above by the normalization conditions (13) (shown by black dashed lines in the figure) and thus the right local maximum at $\alpha \simeq \alpha_0$ cannot overcome the one at $\alpha \simeq 0.1$ even in thermodynamic limit. While it is possible that there is a very slow downward flow of the left local maximum, which will eventually restore GOE, we do not see it within the available system sizes. Further support for possible multifractality at $W = 2.6$ can be obtained by examining the well-known symmetry of multifractal spectrum, which can be analytically derived for wavefunction envelopes in multifractal states of various models [2],

$$f(\alpha) = f(2 - \alpha) + \alpha - 1. \tag{16}$$

While this symmetry is not necessarily satisfied when multifractality is present (for example it fails in localized phases and in some extended phases with Poisson statistics), it serves as an additional indication of multifractality.

In the insets of Fig. 7 we test this symmetry for the maximal available system size $L = 24$. To suppress the effects of zeros of the eigenstates we only examine the symmetry in the regime where the tail of $f(\alpha)$ is significantly above its ergodic value $(3 - \alpha)/2$ (see Eq. (8)), which for our data occurs for $W \geq 2.6$ (see Fig. 6). In this range of disorder strengths the multifractal symmetry (16) is satisfied (insets of Fig. 7), while in the complementary range, $W \leq 2.2$ the symmetry doesn't apply. This indicates a possible multifractal phase for $W > 2.2$.

# 4 Summary and discussion

In this work we have conducted a detailed large-scale numerical study of multifractal properties of eigenstates on the delocalized side of the many-body localization transition (MBL). This phase is known to have a number of anomalous dynamical features, such as subdiffusive transport, sublinear entanglement entropy growth and suppressed spreading of information [22]. For the single-particle case, suppressed relaxation and dynamics are often associated with spatial sparseness of the underlying eigenstates [30, 31]. A natural question to ask is whether a similar relation exists also in the many-body case, namely if sparseness of the eigenstates in Hilbert space implies slow relaxation and suppressed transport of *local* observables. In this work we answer this question in the *negative*, by identifying a large fraction of the delocalized phase which is consistent with ergodicity, while still showing a clear signature of subdiffusion and slow relaxation in both numerical and experimental data. We reach this conclusion by a careful analysis of the finite size flows of eigenstate coefficient distributions, moments of these distributions and their spectrum of multifractal dimensions. Our analysis focuses on the computational basis, where the basis states are labeled by the eigenvalues of the local $\hat{S}_i^z$ operators. This is the natural basis for the XXZ chain in the context of MBL since it is compatible with the disorder and is naturally linked to the hopping problem in Hilbert space. To the best of our knowledge, the multifractal spectrum of the disordered XXZ chain has not been studied before due to severe finite-size behavior and non-monotonic behavior (see Figs. 5 and 6 for example), which hindered reliable extrapolation to the thermodynamic limit.

In this work we focus on standard multifractal probes, namely, on the spectrum of fractal dimensions $f(\alpha)$ and on its Legendre transform, the critical exponent $\tau_q$ of the generalized IPR. We distinguish between mean and typical averaging of $\tau_q$ over different eigenstates and disorder strengths. All the measures we study provide a coherent picture of a steady flow towards the predictions of GOE for disorder strengths $W \lesssim 2.6$. The average $\tau_q$ deviates from the ergodic limit $q-1$ only in the atypical region, $f(\alpha) < 0$, which corresponds to bin counts decreasing with the system size in the wavefunction histogram. At the same time typical $\tau_q$ and multifractal spectrum $f(\alpha)$ demonstrate a district flow towards GOE values. The typical $\tau_q$ show a slight non-concavity, in this disorder interval, due to sub-leading non-linear finite-size effects similarly to the behavior in Anderson localization [2, 74], however this non-concavity is within the error bars and can thus can be safely ignored. A slight discrepancy compared to the GOE prediction, is observed for $W < 1$, as an excess of small values of the eigenstate coefficients compared to GOE. Although it is consistent with a so-called weak ergodic phase, where the wavefunction occupies a finite, but tiny fraction of the Hilbert space, observed in several single-particle and many-body systems [75–77], we argue that this discrepancy is a result of proximity to an integrable point at $W = 0$, and should — if this is indeed the case — disappear either in the thermodynamic limit, or if integrability is broken. We leave the verification of this prediction to a future study.

For larger disorder strengths, our analysis becomes unreliable, due to slowing down of finite-size flows. While we cannot rule out a slow residual flow to GOE (which would provide an alternative explanation in line with strong finite-size effects [56–60]), we don't observe it within our range of accessible system sizes . At this disorder strength, both average and typical $\tau_q$ deviate from their ergodic limit $q-1$ at $q \gtrsim 1$, which is consistent with the saturation of the down-flow of $f(\alpha)$ at $\alpha \lesssim 1$. Moreover, for $W = 2.6$ the multifractal spectrum $f(\alpha)$ perfectly satisfies one of its basic symmetries, which would be consistent with multifractality of the eigenstates in this region. Given the immense numerical cost of our calculations, we could only compute the spectrum in a limited range of disorder strengths across the delocalized phase. Combined with the slow finite size flows at stronger disorder, we cannot determine whether the region consistent with multifractality shrinks to the critical point when the system size is

increased, as was claimed in Ref. [53]. It would be interesting to study this important question in more detail in the future.

One of the central outcomes of our study suggests that the previously observed anomalous dynamics is *not* related to multifractality of many-body eigenstates. However ,since multi-fractal features are generically basis dependent, one can wonder whether the outcome of our study changes with the change of the basis. While it is difficult to predict the effect of a basis rotation without performing actual calculations, it is clear that for GOE eigenstates the multifractal features (i.e. non-fractal in this case) do not depend on the basis, since the GOE distribution is invariant under orthogonal transformation [70]. However, this is only true for *almost all* bases (for example the eigenstate basis is clearly not a good basis to study multifractality). In contrast, for truly multifractal states both in single-particle and many-body systems multifractal and localization properties are drastically basis-dependent. In the Anderson localization community the spatial basis presents a natural choice where the localization transition also show changes in the level statistics [6], but there is no such obvious choice for the many-body case. One good candidate for such a basis, which can be directly tied to relaxation of local observables, is the family of bases generated by *locally* exciting the eigenstates of the system [53].

# Acknowledgments

The authors acknowledge fruitful discussions with Nicolas Macé, Fabien Alet and Nicolas Laflorencie. This research was supported by the Israel Science Foundation (grants No. 527/19 and 218/19). We acknowledge PRACE for awarding access to HLRS's Hazel Hen computer based in Stuttgart, Germany under grant number 2016153659. Our code is based on the PETSC [78, 79], SLEPC [80, 81] and Strumpack [68, 69] libraries. DJL and IMK acknowledge the support from the German Research Foundation (DFG) through SFB 1143 (project ID 247310070) and KH 425/1-1. IMK thanks the support of the Russian Foundation for Basic Research Grant No. 17-52-12044.

# A    Matrix elements

In this appendix, we turn our attention to the analysis of the distributions of matrix elements of the local magnetization $\hat{S}_i^z$ in the eigenbasis of the Hamiltonian. Similarly to our analysis of the eigenstates coefficients in the main text, for each disorder realization we consider 50 eigenstates $|\alpha\rangle$ of the Hamiltonian $\hat{H}$ with an eigenvalue $E_\alpha$ closest to middle of the many-body spectrum $(E_{\max} - E_{\min})/2$. These eigenstates correspond roughly to infinite temperature. We note however, that since we study the microcanonical ensemble with $S_z^{\text{tot}} = 0$ here, where $\text{Tr}_{S_z=0} H \neq 0$, in each disorder realization there is a slightly different "effective temperature", which we correct by subtracting the mean of the diagonal matrix elements (computed over the extracted eigenstates) for each disorder realization (cf. discussion in Ref. [82] and in particular Appendix B therein).

We complement our previous work in Refs. [21, 83], by calculating the distribution of the matrix elements $\langle \alpha | \hat{S}_i^z | \beta \rangle$ of the local $\hat{S}_i^z$ in the eigenbasis $\{|\alpha\rangle\}$ of the Hamiltonian, with a massively improved statistics and one additional system size ($L = 24$). We also add logarithmic binning of the histograms, a direct distribution of diagonal matrix elements rather than their differences as well as a direct comparison of diagonal $\alpha = \beta$ and offdiagonal $\alpha \neq \beta$ matrix

---

[6]Recent developments show that in correlated models the spectrum properties are related to localization in several bases (cf. Ref. [76]).

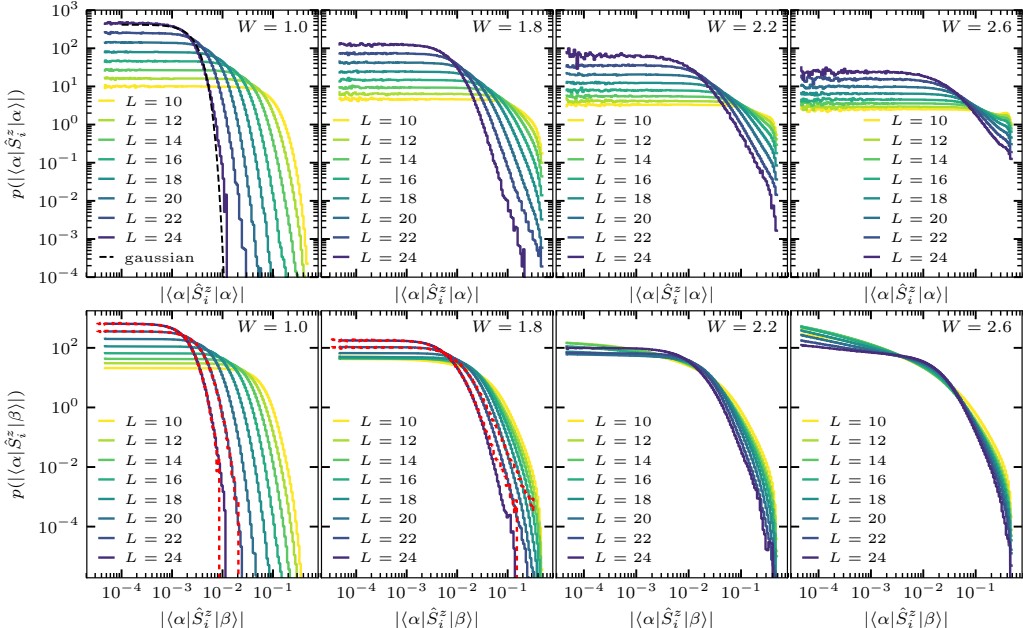

Figure 8: Distribution of diagonal (top) and offdiagonal (bottom) matrix elements of the local $\hat{S}_i^z$ operator as a function of disorder strength $W$ in the eigenbasis $\{|\alpha\rangle\}$ of the Hamiltonian. For each disorder realization, 50 eigenstates closest to middle of the many-body spectrum. For each system size and disorder strength $10^2 \dots 10^4$ disorder realizations are included, as well as all positions $i$ in the chain. Note that for the diagonal matrix elements $\langle\alpha|\hat{S}_i^z|\alpha\rangle$ the distribution for each disorder realization has a (slightly) nonzero mean, which we subtracted here (cf. discussion in Ref. [82]). The red dashed histograms in the bottom row for $W = 1.0$ and $W = 1.8$ correspond to the (rescaled) distribution of the diagonal matrix elements $\sqrt{2}p\left(\langle\alpha|\hat{S}_i^z|\alpha\rangle/\sqrt{2}\right)$ for comparison. For stronger disorder, the distributions of diagonal and offdiagonal matrix elements are so strikingly different that we do not show them in the same panel here.

elements distributions.

Fig. 8 shows the results for the logarithmically binned probability density of diagonal $\left|\langle\alpha|\hat{S}_i^z|\alpha\rangle\right|$ and offdiagonal $\left|\langle\alpha|\hat{S}_i^z|\alpha\rangle\right||$ matrix elements of $\hat{S}_i^z$ for disorder strengths $W = 1.0, 1.8, 2.2, 2.6$, which are well on the delocalized side of the phase diagram, for system sizes $L = 10, 12\dots, 22, 24$. The logarithmic binning highlights the maximum of the distributions, where the matrix elements are closeset to zero. At weak disorder $W \lesssim 1.0$ we note that both diagonal and offdiagonal matrix elements assume a distribution very close to Gaussian as predicted by ETH [84]. Furthermore, the prediction from random matrix theory that distributions of diagonal and offdiagonal matrix elements should be directly related [85, 86] is verified to very high precision (dashed red lines in the lower panels of Fig. 8 are diagonal distributions for $L = 22, 24$ (renormalized by $\sqrt{2}$ in order to take account of convolution of two gaussian distributions) in comparison to offdiagonal distributions shown in color). It is clear however (as was shown in Ref. [86]), that for stronger disorder $W \gtrsim 1.8$, this correspondence is violated.

For the diagonal matrix elements the shape of the maximum appears to be Gaussian (flat on a logarithmic scale), however the tails of the distribution deviate from Gaussian distributions at disorder strengths $W \gtrsim 1.0$. The double logarithmic scale reveals a long straight tail, particularly well developed for $W = 1.8$ and $W = 2.2$, which seems to be consistent with a power law tail over more then one decade.For the offdiagonal matrix elements the tail seems

to decay faster than a power law. In contrast to the diagonal matrix elements, there is a significant excess weight at small values of the offdiagonal matrix elements $\left|\langle\alpha|\hat{S}_i^z|\beta\rangle\right|$, which seems to scale to zero for $W = 2.2$ but survives up to at least $L = 24$ for $W = 2.6$. The scaling of the matrix element distribution variance inversely with Hilbert space dimension is well visible at weak disorder $W = 1.0$ in the (almost) equidistant distributions for both diagonal and offdiagonal matrix elements and was analyzed in detail in Refs. [21, 83]. Increasingly strong deviations from this scaling are observed at stronger disorder, which were connected to subdiffusive transport [21].

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
