# Peer review of "Multifractality and its role in anomalous transport in the disordered XXZ spin-chain"

_SciPost Physics, doi:SciPost Phys. Core 2, 006 (2020)_

## Round 2 · Referee Report · Anonymous (Referee 1) · 2019-11-10

Strengths

1. This work presents a careful large scale numerical study of multifractal properties of the many-body wave function
2. The numerical analysis is carefully executed and authors do not over-interpret their findings, admitting a number of possible scenarios in thermodynamic limit

Weaknesses

1. Work does not provide a sufficient background on multifractality for the reader who is not an expert in this field.
2. In the reverse direction, it would be beneficial to explicitly mention the difference between studies of RRGs and many-body Hamiltonian. The main effect is the correlated nature of the disorder, see e.g. Ref. [arXiv:1901.04384]
3. Work does not have a clear summary of all considered probes and the outcomes of the analysis for each of the probe. Such summary would allow for a much better overview of the results.

Report

The paper by Luitz et al provides a careful numerical study of multifractality in the model used for the numerical studies of the many-body localized phase. They identify a region of disorder strength where mutlifractality is absent. Closer to the MBL transition they find multifractal properties, yet there exists a weak flow towards ergodicity.

The work uses a state-of-the-art numerical data and analyzes it using a number of probes. To the best of my knowledge this is the first study of multifractal spectrum in the many-body system. At the same time, the presentation in this paper can be improved. My overall recommendation is to publish this work after authors improve the presentation.

Requested changes

Following steps would be highly beneficial for improving presentation in the paper:

1. Summary of probes used for studies of multifractality either in the introduction or in the discussion.
2. More detailed discussion of how the fractal properties would change with the change of basis
3. Clear distinction between localization on RRG and MBL models where the disorder is strongly correlated, see Ref. [arXiv:1901.04384].
4. Since authors do not have a space limitation, a more extended review of mutlifractality could improve readability of this paper.
5. It would be good to mention or discuss the form of f(x) for the GOS case somewhere between eq 2-3. Alternatively authors can direct reader to eq 14.
6. Authors mention that tau_q is a concave function. Yet, last column of fig 2 seems to have a slight upward curvature of tau_q for L \in [18,24]. It would be nice to discuss this apparent discrepancy.
7. Last paragraph on page 9 is hard to understand. Please, consider rewriting.
8. It would be nice to explain relation in eq. (16): why does it imply multifractality? Can it be violated in some cases?
9. Consider splitting the first paragraph of Sec. IV. Also this a good place to add the discussion on basis dependence of f(alpha) and clear summary of all considered probes of multifractality and corresponding results. In addition it would be nice to explicitly contrast methods of present study to those of Ref. [52].

Minor comments:

1. Page 2: "Two multifractal moments of the disordered XXZ ..." -- please specify multifractal moments of what quantity are discussed here
2. Space is missing on page 3 "ensemble of matrices[62]"
3. Space is missing on page 5 ".[66]"
4. Please change tau^avg ->tau^{\rm avg} on page 5
5. Why is data for q in range between 3 and 4 not shown?
6. Please explain why different data is needed on page 6 ("for technical reasons we use a different set of data here")
7. "measure - the multifractal" -> "measure --- the multifractal"

  • validity: high
  • significance: good
  • originality: good
  • clarity: low
  • formatting: good
  • grammar: excellent

Author:  Yevgeny Bar Lev  on 2019-12-17  [id 688]

(in reply to Report 1 on 2019-11-10)
Category:
reply to objection

Referee 1

Strengths

  1. This work presents a careful large scale numerical study of multifractal properties of the many-body wave function

  2. The numerical analysis is carefully executed and authors do not over-interpret their findings, admitting a number > of possible scenarios in thermodynamic limit

Weaknesses

  1. Work does not provide a sufficient background on multifractality for the reader who is not an expert in this field.

  2. In the reverse direction, it would be beneficial to explicitly mention the difference between studies of RRGs and many-body Hamiltonian. The main effect is the correlated nature of the disorder, see e.g. Ref. [arXiv:1901.04384]

  3. Work does not have a clear summary of all considered probes and the outcomes of the analysis for each of the probe. Such summary would allow for a much better overview of the results.

Report

The paper by Luitz et al provides a careful numerical study of multifractality in the model used for the numerical studies of the many-body localized phase. They identify a region of disorder strength where mutlifractality is absent. Closer to the MBL transition they find multifractal properties, yet there exists a weak flow towards ergodicity.

The work uses a state-of-the-art numerical data and analyzes it using a number of probes. To the best of my knowledge this is the first study of multifractal spectrum in the many-body system. At the same time, the presentation in this paper can be improved. My overall recommendation is to publish this work after authors improve the presentation.

We thank the referee for the recommendation for publication and for the constructive remarks, which helped us to improve the presentation of our results.

Requested changes

Following steps would be highly beneficial for improving presentation in the paper:

  1. Summary of probes used for studies of multifractality either in the introduction or in the discussion.

A summary of the results has been added to the discussion section.

  1. More detailed discussion of how the fractal properties would change with the change of basis

This question is difficult to answer without performing actual simulations. It is however conceivable that in the case of non-fractal behavior (i.e. when random matrix theory applies), essentially all bases should be equivalent, with a notable exceptions of the eigenbasis. In the case of multifractality, this is less obvious, though we note that the choice of the computational Sz basis is natural in the sense that it is compatible with the eigenbasis of the disorder. We have clarified this point in the discussion.

  1. Clear distinction between localization on RRG and MBL models where the disorder is strongly correlated, see Ref. [arXiv:1901.04384].

While we didn’t intend to review the vast literature on multifractality on RRGs or the Bethe lattice, we have expanded this discussion a bit in the Introduction and highlighted the importance of correlations in the disorder discussed in arXiv:1901.04384 and arXiv:1911.12370.

  1. Since authors do not have a space limitation, a more extended review of mutlifractality could improve readability of this paper.

While indeed there is no space limitation, our work is not aimed to be an introductory work to the topics of multifractality nor MBL. To keep the paper concise we therefore decided to cite the reviews where appropriate. For multifractality we cite the excellent review of Evers and Mirlin which includes all the basics. In the body of our manuscript, we do provide more details on the various multifractal measures and the connections between them.

  1. It would be good to mention or discuss the form of f(x) for the GOS case somewhere between eq 2-3. Alternatively authors can direct reader to eq 14.

We have followed the suggestion of the Referee and redirected the reader to Eq (14) to avoid redundancy.

  1. Authors mention that tau_q is a concave function. Yet, last column of fig 2 seems to have a slight upward curvature of tau_q for L \in [18,24]. It would be nice to discuss this apparent discrepancy.

In the thermodynamic limit tau_q must be concave, the slight non-concavity that the referee mentions is therefore a finite size effect. We have added a discussion of this point to the text.

  1. Last paragraph on page 9 is hard to understand. Please, consider rewriting.

We have rewritten this paragraph

  1. It would be nice to explain relation in eq. (16): why does it imply multifractality? Can it be violated in some cases?

As we explain in the text (16) can be derived analytically for some multifractal models, and it is generically valid for extended states (both ergodic and multifractal) of single-particle disordered models. It however doesn’t say that every multifractal model must satisfy (16) as initially the symmetry is written for the local density of states, but not for the wave function amplitudes. As a result in the localized phases (as well as in some extended phases) with Poisson level statistics (16) may be violated.. Nevertheless, when satisfied, (16) servers as an additional indication of possible multifractality. We have added a clarification of this point around Eq (16).

  1. Consider splitting the first paragraph of Sec. IV. Also this a good place to add the discussion on basis dependence of f(alpha) and clear summary of all considered probes of multifractality and corresponding results. In addition it would be nice to explicitly contrast methods of present study to those of Ref. [52].

We have followed the suggestions of the Referee and split the first paragraph of the discussion, adding the discussion of basis dependence. The methods used in our study and the work by Serbyn et al [52] are in zeroth approximation similar. The details are very technical, and don’t mesh up nicely into the flow of the discussion. Some of the details are provided in Ref. [SciPost Phys. 5, 045 (2018)].

Minor comments:

  1. Page 2: "Two multifractal moments of the disordered XXZ ..." -- please specify multifractal moments of what quantity are discussed here

Thanks, done.

  1. Space is missing on page 3 "ensemble of matrices[62]"

Done

  1. Space is missing on page 5 ".[66]"

Done

  1. Please change tau^avg ->tau^{\rm avg} on page 5

Done

  1. Why is data for q in range between 3 and 4 not shown?

For no particular reason. The departure from the straight line is already seen for q<3, so 3<q<4, doesn’t contain much additional information

  1. Please explain why different data is needed on page 6 ("for technical reasons we use a different set of data here")

Ideally we would like to use the large data set. However, since it was practically impossible to save all the wavefunctions, we had to decide on the analysis at the very beginning of our study. At his stage we didn’t plan to analyze the negative moments, so when we decided to add it later, we had to use a much smaller set, where we did have the eigenstates saved. As the Referee sees, this explanation is purely technical and distracting, so we did not add it to the main text.

  1. "measure - the multifractal" -> "measure --- the multifractal"

Thanks, done

---

## Round 2 · Referee Report · Anonymous (Referee 2) · 2019-12-2

Strengths

The paper addresses an interesting topic, i.e., the role of multifractality of many-body wavefunctions for the dynamics of disordered interacting fermions in a quantum wire at high temperature. The conclusions presented in this work rely on numerical data. The raw data is not shown in the manuscript, but I have no reason to doubt that it has been acquired with state of the art computational technology.

Weaknesses

The weaknesses of this paper, as I see them, fall into three categories: (A) Data analysis; (B) Data interpretation; (C) Embedding into the literature. As will become clear, in this report I can give only a brief illustration of weaknesses. Here I focus on (A) and (B), (C) will be dealt with in the report section.

(A.1) Fig. 2 displays data on the mf-spectrum τq. The analysis assumes a power law, Eq. (1), and adopts an analysis scheme ignoring corrections to scaling, i.e., subleading powers in (1). As is seen from the data in Fig. 2, subleading terms are very large (strong deviations from a straight line). This is the typical situation; methods to deal with it have been developed since the 90ies and are described, e.g., in the review by Evers and Mirlin.
(A.2) The authors' are aware of the fact that τq must be concave. However, their estimate of τq as given in Fig. 3 for W=2.2 does not really conform with this. This is a first indication of a severe problem with the estimated error bars in this work.
(A.3) A second strong indication in the same direction is the fact that typical and average τq are vastly different already at moderate disorder W=2.6 at q-values only slightly larger than unity (Figs. 2,3). Differences are expected, indeed, at moments q high enough; they correspond to (very small) α-values where f(α) takes negative values; such moments are dominated by atypical samples. By inspection of Fig. 6 we see that f(α) turns negative only at α1/3, which is far below the slope α1, at which typical and average depart in Figs. 2,3. Therefore, it is in my mind likely that the apparent discrepancy between typical and average τq reflects an insufficient analysis of error bars.

(B) Without reliable error bars the data presented in the paper cannot be properly interpreted. I expect that a state of the art error analysis will give error bars that resolve the conceptual problems explained in (A). These error bars could be as big as the differences between typical and average τq seen in the present data. They reflect, essentially, the smallness of the current sample and a slow flow towards the true asymptotic regime.
As a consequence, I expect that it will be difficult to extract meaningful asymptotic information from the scaling analysis even after state of the art methods have been applied.

Report

With respect to error analysis, the manuscript should be seen as a typical representative of numerical works that we have seen to appear in the field of MBL in the last couple of years. To illustrate the resulting problems, we mention that the exponent ν of the localization length has been estimated in this way to be ν1, drastically violating the Harris criterion.

A small but growing number of authors is trying to implement a better standard of data analysis, in particular with respect to the effects of system sizes. Their important effect has been clearly worked out, in particular also for the subdiffusive regime that the present authors try to better elucidate with the mf-analysis; see e.g. the works by Weiner et al., Doggen et al..

What I would like to criticize here is that the present authors ignore all these developments in their paper; they present a perspective according to which a set of questions, - e.g. concerning the asymptotic (subdiffusive) dynamics, the value of the critical disorder etc. - can be considered (more or less) settled. This is definitely not the case: I do not see a real consensus in the community. A very recent publication by Panda et al. 1911.07882v1 illustrates this situation. Therefore, I would say that the present paper is in its perspective of the field and its current status highly selective, biased and hence misleading.

Smaller remarks:
- I could not find the definition of the basis |n that has been used for the definition of Fock-space. It is called a "certain" basis or "computational" basis, but other than that I could not find the specification. To be explicit here is important, because one expects the mf-spectrum to depend on the choice of that basis.
- The authors extract 50 states per sample. One would expect that the spectrum to some extent depends on this parameter (due to rounding errors etc.). No evidence to the opposite has been given.

  • validity: low
  • significance: low
  • originality: ok
  • clarity: poor
  • formatting: reasonable
  • grammar: good

Author:  Yevgeny Bar Lev  on 2019-12-17  [id 689]

(in reply to Report 2 on 2019-12-02)

Strengths The paper addresses an interesting topic, i.e., the role of multifractality of many-body wavefunctions for the dynamics of disordered interacting fermions in a quantum wire at high temperature. The conclusions presented in this work rely on numerical data. The raw data is not shown in the manuscript, but I have no reason to doubt that it has been acquired with state of the art computational technology.

We thank the referee for the evaluation of our work. Unfortunately, it is completely impractical to show a few tens of TB of data in the manuscript. Therefore, histograms of this raw data are shown in terms of f(alpha) [Figs. 6 and 7] and wavefunction distributions themselves [Fig. 1] for each system size and disorder strength.

Weaknesses The weaknesses of this paper, as I see them, fall into three categories: (A) Data analysis; (B) Data interpretation; (C) Embedding into the literature. As will become clear, in this report I can give only a brief illustration of weaknesses. Here I focus on (A) and (B), (C) will be dealt with in the report section.

We respectfully disagree with the Referee on all three points and provide a detailed reply to each point of his/her criticism below.

(A.1) Fig. 2 displays data on the mf-spectrum tau_q. The analysis assumes a power law, Eq. (1), and adopts an analysis scheme ignoring corrections to scaling, i.e., subleading powers in (1). As is seen from the data in Fig. 2, subleading terms are very large (strong deviations from a straight line). This is the typical situation; methods to deal with it have been developed since the 90ies and are described, e.g., in the review by Evers and Mirlin.

We assume that the Referee meant Eq. (4) defining the generalized IPR and not Eq. (1) describing the model. We are aware of the power-law subleading corrections mentioned in the review by Evers and Mirlin, but there are two issues with this in the present context. First, the analysis resulting in power-law corrections with irrelevant exponent y developed in 90ies is based on the fact of concavity of f(alpha) which is not the case in our data, see Figs. 6 and 7. We note that this is due to the fact that we do not calculate f(alpha) by a mere inverse Legendre transform but calculate it directly from its definition in Eq. (3) as a logarithmically binned histogram of wavefunction amplitudes. This approach is direct and allows to unveil additional details in f(alpha), which would be lost otherwise.

As a result the prefactor c_q in front of the scaling N^{-\tau_q} in the generalized IPR can be in principle ln(N)-dependent. Secondly, it is clear that subleading power-law terms with irrelevant exponents in (4) cannot provide any significant corrections to tau_q at the considered Hilbert space dimensions ~2^{20-24} compared to other corrections. Indeed, in the above mentioned case of non-concave f(alpha) the main deviations from the straight line of tau_q(N) versus 1/ln(N) come from the logarithmic prefactors in front of the scaling of the generalized IPR (giving ln[ln(N)]/ln(N) corrections in tau_q instead of ln(1+N^{-y})/ln(N) which comes from subleading powers). These finite-size corrections are known to be more significant at large moments q, where the main contribution to tau_q is given by the tails of the distribution of alpha = -ln |psi|^2/ln(N) given by f(alpha). This is one of the reasons we restrict our analysis to q<3. Another possible source of deviations in tau_q(N) is the mixture of states with different multifractal dimensions (for example, across the mobility edge). In order to get rid of this in our analysis we compare the results of average and typical tau_q and show that the latter, which is less sensitive to the variation between states, is much closer to the ergodic limit tau_q = q-1. These aspects related with non-concave f(alpha) complete the analysis done in 90ies and mentioned in the review by Evers and Mirlin. It is used in growing number of papers and should become standard.

(A.2) The authors' are aware of the fact that tau_q must be concave. However, their estimate of tau_q as given in Fig. 3 for W=2.2 does not really conform with this. This is a first indication of a severe problem with the estimated error bars in this work.

The concavity of tau_q is the property of the Legendre transform and it works for all finite-size curves tau_q(N). The non-convacity in some plots (like W=2.2 in Fig. 3) is within the error bars as is clearly seen from the figure. We do not see any problems with error bars estimated with a gold standard bootstrap procedure tracing back the error through the entire data analysis chain, including blocking to decorrelate samples.

(A.3) A second strong indication in the same direction is the fact that typical and average tau_q are vastly different already at moderate disorder W=2.6 at q-values only slightly larger than unity (Figs. 2,3). Differences are expected, indeed, at moments q high enough; they correspond to (very small) α-values where f(α) takes negative values; such moments are dominated by atypical samples.

The comment of the Referee is based on the discussion of typicality described for the spectrum of fractal dimensions in the review by Evers and Mirlin. Indeed, the typical f(alpha) coincides with the average one above the x-axis. However it works only for the states with homogeneous multifractal spectrum. If there is a small fraction of different states, the definition of typical should be modified. In general, the typical averaging of tau_q do not correspond to the typical spectrum of fractal dimensions. Another aspect is again related to non-concavity of f(alpha) as for the Legendre transform calculation the non-concave parts of f(alpha) do not contribute to tau_q and thus the effective “cutoff” of typical f(alpha) may occur at positive f(alpha).

By inspection of Fig. 6 we see that f(α) turns negative only at α≲1/3, which is far below the slope α∼1, at which typical and average depart in Figs. 2,3. Therefore, it is in my mind likely that the apparent discrepancy between typical and average tau_q reflects an insufficient analysis of error bars.

In Fig. 6 for W=2.6 The slope at the point alpha~⅓ where f(alpha)=0 is about df/dalpha~1/(1-alpha) = 1.5 which corresponds to the value of q = df/dalpha at which both average and typical tau_q start to deviate from q-1 in full agreement with the above mentioned review. The origin of the Referee’s confusion is in the fact that f(alpha) is non-concave already at finite sizes (this case is not covered by the above mentioned review as in most of the works f(alpha) is approximated by the inverse Legendre transform of tau_q, here we use directly it’s exact definition in Eq. (3) ). As a result, the slope of tau_q should change abruptly in the thermodynamic limit from the value dtau_q/dq = alpha_q close to 1 to the value close to ⅓ mentioned by the Referee. All of this is consistent in our data and analysis as it is just follows from the properties of the Legendre transform.

(B) Without reliable error bars the data presented in the paper cannot be properly interpreted. I expect that a state of the art error analysis will give error bars that resolve the conceptual problems explained in (A). These error bars could be as big as the differences between typical and average tau_q seen in the present data. They reflect, essentially, the smallness of the current sample and a slow flow towards the true asymptotic regime. As a consequence, I expect that it will be difficult to extract meaningful asymptotic information from the scaling analysis even after state of the art methods have been applied.

In all figures, except Fig. 7 the error bars are explicitly shown. The error bars are much smaller than the differences between typical and average tau_q and they allow to distinguish these two probes. We use a standard Monte Carlo resampling technique (bootstrap)to calculate reliable error bars. This method is standard in both Monte-Carlo and high energy communities. All this is clearly stated in the text. The system size is as big as 24 interacting spins, which is at the top edge of the possible sample sizes achieved by state-of-the-art diagonalization techniques (Hilbert space dimension 2.7 million) for the calculation of interior eigenpairs. The error bars are shown everywhere except Fig. 7 (for clarity) which presents the same data as Fig. 6. The error bars in Fig. 6 are shown by filled region, but so small that can be seen only as thicker lines at the tails. We have clarified it in all captions.

Report With respect to error analysis, the manuscript should be seen as a typical representative of numerical works that we have seen to appear in the field of MBL in the last couple of years. To illustrate the resulting problems, we mention that the exponent nu of the localization length has been estimated in this way to be nu≈1, drastically violating the Harris criterion.

We would like to respectfully note that critical exponents as well as the perceived quality of other works in the field are irrelevant for the assessment of the current manuscript. We would therefore appreciate if the Referee could raise a concrete statement.

A small but growing number of authors is trying to implement a better standard of data analysis, in particular with respect to the effects of system sizes. Their important effect has been clearly worked out, in particular also for the subdiffusive regime that the present authors try to better elucidate with the mf-analysis; see e.g. the works by Weiner et al., Doggen et al..

The time-dependent variational principle used in some of the recent works by Doggen et al. is not applicable to our objective, which relies on high precision exact central eigenvectors of the Hamiltonian in the full phase diagram. We agree with the referee that it is significantly easier to calculate real time dynamics of wave functions compared to exact eigenvectors and much larger system sizes can be reached, as some of the authors have demonstrated in other works in the context of MBL and Floquet systems, cf. the review [Annalen der Physik 529, 1600350 (2017)] for a discussion of the relevant methods. While eigenstates can be quite reliably extracted using MPS based techniques in the localized region [PRL 116, 247204; PRL 118, 017201 ], they fail in the delocalized and critical regions, which we study in the present work.

What I would like to criticize here is that the present authors ignore all these developments in their paper; they present a perspective according to which a set of questions, - e.g. concerning the asymptotic (subdiffusive) dynamics, the value of the critical disorder etc. - can be considered (more or less) settled. This is definitely not the case: I do not see a real consensus in the community. A very recent publication by Panda et al. 1911.07882v1 illustrates this situation. Therefore, I would say that the present paper is in its perspective of the field and its current status highly selective, biased and hence misleading.

The presence of the MBL transition in XXZ Heisenberg chain is accepted by the community, though we don’t see the relevance of this to our work . The critical disorder strength is also more or less settled (with a large uncertainty). Of course the finite-size effects are significant in many-body systems (as well as in hierarchical single-particle ones like RRG) and it is demonstrated in several recent papers including 1911.07882, 1911.04501, 1911.06221. In our work we decided to avoid using any uncontrollable approximations and apply state-of-the-art exact diagonalization techniques. In the new version of the manuscript we add a discussion of the above mentioned recent papers. In the work we present all the raw data, and pursuing a standard multifractal analysis, which is by no means biased. Our interpretation of the data, which is mostly limited to the discussion section is based on at least two possible scenarios. We therefore do not understand in which sense it is misleading. In any case, readers can choose their own interpretation of the raw data and the analysis that we provide.

Smaller remarks: - I could not find the definition of the basis |n⟩ that has been used for the definition of Fock-space. It is called a "certain" basis or "computational" basis, but other > than that I could not find the specification. To be explicit here is important, because one expects the mf-spectrum to > depend on the choice of that basis.

The computational basis or sigma-z basis is a standard notation in the MBL community for spin models. We add the definition of the basis into the manuscript.

  • The authors extract 50 states per sample. One would expect that the spectrum to some extent depends on this parameter (due to rounding errors etc.). No evidence to the opposite has been given.

We do not understand how the rounding errors depend on the number of eigenstates extracted per sample. This being said, we have explicitly ensured that our numerical estimate of generalized IPR is not hampered by cancellation errors using two independent strategies: compensated summation and arbitrary precision arithmetic. This analysis revealed that double precision arithmetic yields identical results. We take 50 eigenstates in the middle of the spectrum which corresponds to the infinite temperature states. Due to the presence of 10^2 ... 10^4 disorder realizations and 10^5...2.7 10^6 wave function coefficients for each eigenstate, the sample size of f(alpha) and tau_q calculations is enormous. Selecting few (50) states per sample allows us to minimize correlation in the data, while increasing the total sample size significantly. In the limit of enormous statistics of the data nothing will depend on the number of states in the sample. On the contrary, it may depend on the product of the number of states per sample, the number of disorder realizations and the Hilbert space dimension.

---

## Round 3 · Referee Report · Anonymous (Referee 2) · 2020-2-14

Report

My answer to the authors’ reply I would like to keep brief. In essence, I am not satisfied with the reply because it does not address the main point of my criticism: I do not believe that the multifractal scaling analysis presented in this work gives exponents with an accuracy as claimed by the authors. Instead, the material presented in the manuscript is, in my opinion, inconclusive. For instance, the summary makes a claim: “One of the central outcomes of our study suggests that the previously observed anomalous dynamics is not related to multifractality of many-body eigenstates”. In my mind, the claim has no clear support from the presented material.

Furthermore, I consider the manuscript misleading because the dramatic impact of finite-size corrections is not sufficiently accounted for in the paper’s conclusions. The authors say “To the best of our knowledge, the multifractal spectrum of the disordered XXZ chain has not been studied before due to severe finite-size behavior […], which hindered reliable extrapolation […].” What the authors don’t say is why to believe that their system sizes (up to L=24) are significantly better than earlier ones (L=18,20,22).

Finally, I believe that the presentation of the status of the field MBL as offered in this manuscript is selective and biased. Evidence indicating the importance of finite-size effects have been reported in other recent studies, e. g. by the Princeton group, see below. As far as I can see, this evidence is ignored in the review authored by Luitz and Bar Lev, Ref. [23], cited in the summary; they are also ignored in the present manuscript.

Summarizing, in my opinion the manuscript should not be published.

I repeat the observations underlying my recommendation: The authors have a spread in system sizes, L=10-24, i.e. by only a factor of roughly two. The common belief – and also my own experience – is that usually a spread in system sizes by (at least) two orders of magnitude is required in order to safely identify a power law and the corresponding exponent in critical localization behavior. While this statement does not , in principle, exclude the possibility of exceptions, in practice strong indications exist in the data presented in this (and earlier) work that exclude the possibility that the disordered XXZ-model is of such an exceptional sort.

(1) Immediate evidence for this claim: Strong finite size effects are observed in the data, see e.g. Fig. 2 at W=2.2 and 2.6. In their data analysis the authors do not account for these effects; subleading corrections to scaling are not included in the fitting. Therefore, the conclusions may be seriously impaired by the latter.

(2) Indirect evidence: (a) The present paper reports that typical and average values for exponents do not coincide. Such a finding is hard to reconcile with conventional wisdom; it is, however, in my experience a typical indicator of finite-size corrections that have not been accounted for, properly. (b) Similarly, f(\alpha) is reported to be non-convex, see Fig. 6. Also this behavior is contradicting established knowledge and, usually, emphasizes the importance of corrections to scaling.

(3) Circumstantial evidence: The authors agree in their response that the critical point in the model they investigated is not well known. The value cited in the text, W_c\sim 3.7, has been questioned, e.g. by Devakul and Singh, who give a much larger value W_c \approx 4.5 as a lower bound (PRL 2015). The issue is significant for the present case because it demonstrates the importance of corrections to scaling; the asymptotic behavior in this model is just not under computational control.

I add that subsequent analysis of Khemani, Sheng and Huse (PRL 2017) corroborates this conclusion. In fact, these authors recommend explicitly to study MBL in quasiperiodic systems rather than in the random systems considered by the present authors, because “the entanglement structure at the critical fixed points in RG studies [..] indicates that the asymptotic disorder dominated regime in these random models might only be apparent in samples larger than ~100 spins [..], [..].” The importance of finite size effects has been emphasized again also by other authors and, in particular, also for the range of disorder values considered in the present manuscript. (e.g. Weiner et al.; Doggen et al. ).
  • validity: -
  • significance: -
  • originality: -
  • clarity: -
  • formatting: -
  • grammar: -

Author:  Yevgeny Bar Lev  on 2020-04-02  [id 786]

(in reply to Report 1 on 2020-02-14)
Category:
answer to question
reply to objection
pointer to related literature

**Referee**

My answer to the authors’ reply I would like to keep brief. In essence, I am not satisfied with the reply because it does
not address the main point of my criticism: I do not believe that the multifractal scaling analysis presented in
this work gives exponents with an accuracy as claimed by the authors. Instead, the material presented in the
manuscript is, in my opinion, inconclusive. For instance, the summary makes a claim: “One of the central outcomes of
our study suggests that the previously observed anomalous dynamics is not related to multifractality of many-body
eigenstates”. In my mind, the claim has no clear support from the presented material.

**Our reply**

We thank the referee for the second evaluation of our work. The performed analysis meets the high standards of the Monte-Carlo and high energy physics communities. We have used a state of the art bootstrap (Monte Carlo resampling) analysis for all results shown in the manuscript, tracing back the error through the entire data analysis chain and removing correlations by blocking. We are therefore confident that the non-systematic error bars shown in the manuscript are reliable. As is always the case for multifractal studies the data is more reliable for smaller (in magnitude) moments, and our data is not different in this respect. We have strong evidence of ergodicity for smaller q-s, but the flow to ergodicity for larger q-s is less conclusive. A point, which we clearly discuss in the manuscript.

**Referee**

Furthermore, I consider the manuscript misleading because the dramatic impact of finite-size corrections is not
sufficiently accounted for in the paper’s conclusions. The authors say “To the best of our knowledge, the multifractal
spectrum of the disordered XXZ chain has not been studied before due to severe finite-size behavior […], which
hindered reliable extrapolation […].” What the authors don’t say is why to believe that their system sizes (up to L=24)
are significantly better than earlier ones (L=18,20,22).

**Our reply**

As we say in the text, we are not aware of previous studies of the multifractal spectrum of the XXZ model, therefore we are not sure what the Referee means when he asks us to compare our system sizes to the “earlier ones”. As can be seen from our results, there is a significant curving in the data even for the lower moments, for system size up-to L=16, 18, which is roughly the limit of exact diagonalization.
Of course one cannot exclude other sources of the finite-size effects, but in the extrapolation of multifractal measures (critical exponents tau_q and the spectrum of fractal dimensions f(alpha)) the Hilbert space dimensions plays the role of the extrapolating parameter (the finite size corrections are proportional to 1/ln N, but not 1/ln L). As the Hilbert space dimension spreads in our work from 252 to 2 704 156 (which makes 104 times difference) which is at the maximal range achieved with exact diagonalization methods so far. Going to L=24 from L=22, increases the Hilbert space by a factor of 4 compared to L=22, which is not a negligible increase. One can compare to similar efforts done for the 3D Anderson model for example.

**Referee**

Finally, I believe that the presentation of the status of the field MBL as offered in this manuscript is selective and
biased. Evidence indicating the importance of finite-size effects have been reported in other recent studies, e. g. by
the Princeton group, see below. As far as I can see, this evidence is ignored in the review authored by Luitz and Bar
Lev, Ref. [23], cited in the summary; they are also ignored in the present manuscript.

**Our reply**

While we are not sure why the Referee uses this stage to discuss his/her dissatisfaction with a different manuscript that two of us authored, we have included the discussion of finite-size effects in the XXZ model that the Referee mentions. We however feel that the relevance is tangential, since these studies discuss finite-size effects in the context of dynamical quantities and not multifractality, which are not necessarily related.

**Referee**

Summarizing, in my opinion the manuscript should not be published.

I repeat the observations underlying my recommendation: The authors have a spread in system sizes, L=10-24, i.e. by
only a factor of roughly two. The common belief – and also my own experience – is that usually a spread in system
sizes by (at least) two orders of magnitude is required in order to safely identify a power law and the corresponding
exponent in critical localization behavior. While this statement does not , in principle, exclude the possibility of
exceptions, in practice strong indications exist in the data presented in this (and earlier) work that exclude the
possibility that the disordered XXZ-model is of such an exceptional sort.
(1) Immediate evidence for this claim: Strong finite size effects are observed in the data, see e.g. Fig. 2 at W=2.2 and
2.6. In their data analysis the authors do not account for these effects; subleading corrections to scaling are not
included in the fitting. Therefore, the conclusions may be seriously impaired by the latter.

**Our reply**

It is likely that the experience of the Referee lies mostly with non-interacting systems, where indeed system sizes with a linear dimension of about L=200 were required to reliably perform the multifractal analysis [10.1103/PhysRevLett.105.046403]. This however corresponds to a Hilbert space dimension of L^3 =10^6. We would like to point out that the Hilbert space dimension of the L=24 system is 2,704,156, which is of the same order of magnitude used for the Anderson transition.

**Referee**

(2) Indirect evidence: (a) The present paper reports that typical and average values for exponents do not coincide.
Such a finding is hard to reconcile with conventional wisdom; it is, however, in my experience a typical indicator of
finite-size corrections that have not been accounted for, properly. (b) Similarly, f(\alpha) is reported to be non-
convex, see Fig. 6. Also this behavior is contradicting established knowledge and, usually, emphasizes the importance
of corrections to scaling.

**Our reply**

The common lore behind the concavity of f(alpha) comes from its calculation in terms of the inverse Legendre transform, which is able to reconstruct only the concave hull of f(alpha) from the critical exponents tau_q (see, e.g., Touchette Phys.Rep. 478, 1 (2009) for the review on the inverse Legendre transform). However this view is somehow dated, since already 5 years ago the first example of non-convex spectrum of fractal dimensions f(alpha) was found in Ref. [42] and rigorously proven mathematically in 2018 [P. von Soosten and S. Warzel, Letters in Mathematical Physics 109(4), 905 (2019)]. This example was followed by several other random-matrix [68] and many-body models [Micklitz et al PRL 123, 125701 (2019), Faoro et al. Annals of Physics 409, 167916 (2019)] showing similar behavior of f(alpha).
In this work (as well as in growing body of recent works) the spectrum of fractal dimensions has been calculated directly from the distribution function of the wave function coefficients approximated a histogram of wave function amplitudes with logarithmic bins alpha = -ln |psi|^2 / ln N.

Following from this non-concavity of f(alpha), the effective q where tau_q starts to deviate from its ergodic behavior tau_q = q-1 may correspond to the positive f(alpha) values at the smallest alpha where f(alpha) is still concave as around the maximum alpha = alpha_0.

**Referee**

(3) Circumstantial evidence: The authors agree in their response that the critical point in the model they investigated
is not well known. The value cited in the text, W_c\sim 3.7, has been questioned, e.g. by Devakul and Singh, who give
a much larger value W_c \approx 4.5 as a lower bound (PRL 2015). The issue is significant for the present case
because it demonstrates the importance of corrections to scaling; the asymptotic behavior in this model is just not
under computational control.

I add that subsequent analysis of Khemani, Sheng and Huse (PRL 2017) corroborates this conclusion. In fact, these
authors recommend explicitly to study MBL in quasiperiodic systems rather than in the random systems considered by
the present authors, because “the entanglement structure at the critical fixed points in RG studies [..] indicates that
the asymptotic disorder dominated regime in these random models might only be apparent in samples larger than
~100 spins [..], [..].” The importance of finite size effects has been emphasized again also by other authors and, in
particular, also for the range of disorder values considered in the present manuscript. (e.g. Weiner et al.; Doggen et al.
).

**Our reply**

We hardly see the relevance of the discussion of the critical disorder strength to our point. All the works in the literature agree that the critical disorder is at least W_c = 3.7, while some works mentioned by the Referee and by us in the previous reply claim W_c ~ 4.5 or higher. In the current work we deliberately stayed within the presumably ergodic phase, by choosing the disorder strength W<=3.0, which is far below all the estimates of the transition point. Moreover we suggest that our data can be consistent with multifractal behavior already at W = 2.6 which is at least by 30 - 40 % below the estimated critical disorder. As we mentioned above, we have added the discussion of finite-size effects to the beginning of the Model section.

---

## Round 3 · Referee Report · Anonymous (Referee 1) · 2020-2-27

Report

After studying the report of another referee and authors' reply I see two different approaches. The authors infer error bars and conclusions based solely on data and on scaling relations for conventional multifractal systems. This is fair approach given, that there exists no proper theory for MBL systems. At the same time, the referee keeps emphasizing the severe nature of finite size effects and concludes that multifractality is a signature of the critical region. The extremely large critical region is indeed supported by the growing body of work, however is not rigorously established either.

Hence, I suggest that the authors and referee can meet each other "in between": In the v3 of their paper authors already implemented a number of changes to acknowledge strong nature of finite size effects. I suggest them to do it more explicitly, suggesting that a different interpretation is possible. In addition to the parts of the main text that discuss deviation from the scalings, it would be nice to include such acknowledgment also in:

$\bullet$ Introduction, right after sentence "Our analysis thus allows us to locate a region in the extended phase which appears to be nonergodic." one can explain that it is not possible to rule out strong finite size effects citing refs 57-59.

$\bullet$ In the discussion instead of writing "While we cannot rule out a slow residual flow to GOE..." authors can say that this would provide an alternative explanation in lines with strong finite size effects suggested in Ref 57.

$\bullet$ When authors conclude that mutlifractality $\neq$ anomalous transport they should be careful since (as they write themselves) mutifractality can be strongly basis dependent.

Finally, let me reiterate the basis for my recommendation to accept this work. In my view it does address an important question using state of the art method. I do not know any other instances of the fractal spectrum studies for many-body systems. Observation of fractal spectrum symmetry is also an interesting finding that calls for an explanation. Provided authors stay more open on the interpretation of the numerically exact data, this work provides a non-trivial addition to the literature.
  • validity: -
  • significance: -
  • originality: -
  • clarity: -
  • formatting: -
  • grammar: -

Author:  Yevgeny Bar Lev  on 2020-04-02  [id 787]

(in reply to Report 2 on 2020-02-27)
Category:
correction

**Referee**

After studying the report of another referee and authors' reply I see two different approaches. The authors infer error
bars and conclusions based solely on data and on scaling relations for conventional multifractal systems. This is fair
approach given, that there exists no proper theory for MBL systems. At the same time, the referee keeps emphasizing
the severe nature of finite size effects and concludes that multifractality is a signature of the critical region. The
extremely large critical region is indeed supported by the growing body of work, however is not rigorously established
either.

**Our reply**
We thank the referee for the careful reading of our work, the other Referee reports and our replies. We also appreciate the constructive remarks of the Referee, which helped us to improve the presentation of our results.

**Referee**

Hence, I suggest that the authors and referee can meet each other "in between": In the v3 of their paper authors
already implemented a number of changes to acknowledge strong nature of finite size effects. I suggest them to do it
more explicitly, suggesting that a different interpretation is possible. In addition to the parts of the main text that
discuss deviation from the scalings, it would be nice to include such acknowledgment also in:

Introduction, right after sentence "Our analysis thus allows us to locate a region in the extended phase which appears
to be nonergodic." one can explain that it is not possible to rule out strong finite size effects citing refs 57-59.∙
In the discussion instead of writing "While we cannot rule out a slow residual flow to GOE..." authors can say that this
would provide an alternative explanation in lines with strong finite size effects suggested in Ref 57.∙

**Our reply**

We thank the referee for the suggestion and follow it in the new version of the manuscript.

**Referee**

When authors conclude that mutlifractality ≠ anomalous transport they should be careful since (as they write
themselves) mutifractality can be strongly basis dependent.
We have made a number of changes, which makes this point more clear, in the last paragraph of the discussion.

Finally, let me reiterate the basis for my recommendation to accept this work. In my view it does address an
important question using state of the art method. I do not know any other instances of the fractal spectrum studies
for many-body systems. Observation of fractal spectrum symmetry is also an interesting finding that calls for an
explanation. Provided authors stay more open on the interpretation of the numerically exact data, this work provides
a non-trivial addition to the literature.

**Our reply**

We thank the Referee for the recommendation to accept our work and hope that the current version is more open on the interpretations.

---

## Editorial Decision

published